# Effect of Polystyrene Microplastics on the Antioxidant System and Immune Response in GIFT (*Oreochromis niloticus*)

**DOI:** 10.3390/biology12111430

**Published:** 2023-11-14

**Authors:** Yao Zheng, Tracy Naa Adoley Addotey, Jiazhang Chen, Gangchun Xu

**Affiliations:** 1Key Laboratory of Integrated Rice-Fish Farming Ecology, Ministry of Agriculture and Rural Affairs, Freshwater Fisheries Research Center (FFRC), Chinese Academy of Fishery Sciences (CAFS), No. 9 Shanshui East Rd., Wuxi 214081, China; zhengy@ffrc.cn (Y.Z.); chenjz@ffrc.cn (J.C.); 2Wuxi Fishery College, Nanjing Agricultural University, No. 9 Shanshui East Rd., Wuxi 214081, China; tracy.adoley@gmail.com

**Keywords:** microplastics, antioxidant, *Chlorella*, reactive oxygen species, immune, inflammation

## Abstract

**Simple Summary:**

This study determined the levels of IL-1ß, TNF-α, ROS, and SOD in the brain, gills, liver, and intestine of tilapia exposed to different sizes of polystyrene microsphere microplastics (MPs) with or without the existing *Chlorella*. MPs may induce excessive ROS production and affect the antioxidative enzyme system, with the highest observed in the brain being 750 μm. A total of 750 μm of MPs may cause the over expression of brain IL-1ß and TNF-α. *Chlorella* may suffer oxidative stress caused by the presence of MPs.

**Abstract:**

Recent studies have revealed a significant presence of microplastics (MPs) in freshwater ecosystems, raising concerns about their potential negative impacts on the growth and development of freshwater organisms. The present study was conducted to examine the effects of chronic sub-lethal doses of polystyrene microsphere MPs on the oxidative status (ROS, SOD) and the immune response (IL-1ß, TNF-α) of genetically improved farmed tilapia (a kind of tilapia hereafter referred to as GIFT). GIFT juveniles (5.1 ± 0.2 g) were exposed to different concentrations of substances. The experimental groups were as follows: group A (control, no exposure), group B (exposed to a concentration of 75 nm), group C (exposed to a concentration of 7.5 μm), group D (exposed to a concentration of 750 μm), group E (exposed to a combination of 75 nm, 7.5 μm, and 750 μm), and group F (exposed to a combination of 75 nm and *Chlorella*). The ROS contents in the brain and gills were significantly decreased in group F, while a significant increase was observed in group D following a 14-day exposure. SOD activities in the intestine showed an elevation in group F, as did those in the brain and gills in group D, while the SOD levels in the gills generally decreased over time in groups B and F. Notably, the highest ROS and SOD were observed in the brain of group D, whereas the lowest were in the intestines at the same concentration. The activity of IL-1β in the liver was significantly up-regulated in all of the exposure groups. IL-1β was significantly up-regulated in the brain of group B and in the gills of group D. Similarly, TNF-α was significantly up-regulated in the brain of groups B/D/E, in the liver of groups B/C/D, in the intestine of group B, and in the gills of group D. Notably, the highest levels of IL-1β and TNF-α activities were recorded in the brain, while the lowest were recorded in the intestine of group D. Overall, this study revealed that GIFT’s immune response and antioxidant system can be affected by MPs.

## 1. Introduction

Microplastics (MPs) refer to plastic particles with a diameter of less than 5 mm that can accumulate in aquatic water bodies and sediments and are consumed by a variety of aquatic species [1]. The presence of microplastics (MPs) in aquaculture has been observed to be on the rise, and the pollution caused by MPs in freshwater systems is as serious as that in the oceans in China. MPs’ abundancies in Dongting Lake and Hong Lake ranged from 900 to 2800 and 12,504,650 items per m^−3^, respectively [2], while in Taihu Lake, the abundance of MPs in the surface water ranged from 3.4 to 25.8 items per L^−3^ [3]. There is growing concern about the effects of MPs on aquatic habitats due to the necessity for aquatic organisms to discriminate between natural food sources and sources of MPs [4].

MPs are predominantly accumulated in the intestinal cavities of fish that have been exposed to them. Once ingested, these MPs are absorbed and can metastasize in tissues and the hemolymph [5]. To avoid duplication of MPs, concentrations in various tissues were increased over a period of time following the order gut > gills > liver > brain during a 14-day exposure to 1.1 mm MPs at concentrations of 1, 10, and 100 mg L^−1^ in red tilapia (*Oreochromis niloticus*) [6]. Similarly, accumulation of MPs was observed in the gills, liver, and gut of zebrafish (*Danio rerio*) after a 7-day exposure to 20 mg L^−1^ [7]. MPs ingested by fish can block their feeding appendages, obstruct food passage, or trigger pseudo satiation, ultimately resulting in diminished food consumption [8]. MPs with diameters ranging in the tens of micrometers have the propensity to translocate from the gut to the circulatory system in several fish species, where they may stay for extended durations. Ingestion of MPs by aquatic fish species has been found to elicit inflammatory responses, which may lead to negative effects such as oxidative damage, decreased energy reserves, reduced reproduction, metabolic interference and cellular lesions [1]. Oxidative stress refers to the imbalance state that occurs when there is an excessive production of reactive oxygen species (ROS) and the resulting impact on the cells’ ability to reduce ROS levels. Based on the concept of defense mechanisms, antioxidants can be classified into three categories: first-line defense, represented by SOD; second-line defense, represented by CAT; and third-line defense, represented by GPx. The superoxide dismutate radical is responsible for the decomposition of hydrogen peroxides and hydroperoxides to non-toxic molecules (H_2_O_2_/alcohol and O_2_). GSH reduces H_2_O_2_ and lipid hydroperoxides through the action of GPx enzymes. This process is an important part of the integrated antioxidant system and of ensuring the maintenance of other non-protein antioxidants in their reduced and biologically significant state. MPs have been found to cause histological deviations and oxidative stress in the gut of zebrafish. [9]. Recent research findings have shown that microplastics can inhibit growth and induce oxidative damage in the liver, intestine, and gills of crucian carp [10]. MPs can be ingested by fish and cause adverse effects such as oxidative stress [7,10]. Tumor necrosis factor alpha (TNF-α) and interleukin-1β (IL-1β) found in fish are commonly used by researchers as biomarkers for fish immunity. TNFα can induce either NF-kB-mediated survival or apoptosis, while IL-1β is a potent pro-inflammatory cytokine that plays a critical role in the host’s defense responses to infection and injury. MPs (5 μm and 70 nm) with a 2000 μg L^−1^ exposure concentration have been found to induce inflammation in the intestine [11] and the accumulation of lipid in the liver, an indicator that MPs may cause pathological symptoms in zebrafish [7].

Research findings have proven that MPs in fish tissues should be of particular concern [12], because the consumption of contaminated foods including seafood may serve as a medium for MPs to be ingested and translocated to humans [13]. Microalgae have a strong propensity for adsorbing pollutants and retaining them in the microalgae phase [14], especially for MPs [15]. *Chlorella pyrenoidosa* is one of the most widely distributed and common types of algae in freshwater environments, and it has been employed recently as a model organism in toxicological research due to its special features, such as short generation time and reactivity to contaminants [16]. *C. pyrenoidosa* is sensitive to environmental threats; therefore, understanding the interactions between the microalgae (*Chlorella*) and MPs is key to assessing the evolving impact of MPs on the ecological properties in aquatic environments. MPs have been proven to cause direct physical injury and structural deviations in algae [17] as well as oxidative stress via free radical generation [18]. As a result, there is an urgent need to investigate the bioaccumulation potential of MPs in freshwater fishes. This investigation has been recently applied to monitor alterations in the aquatic ecosystem and to assess the ecological risks caused by MPs pollution [19]. However, there is a lack of comprehensive data on the bioaccumulation of MPs in freshwater fish, especially at the tissue level, and most importantly, the effects on the health and metabolism of fish tissues cannot be ignored. The primary objective of this study is to evaluate the effects of polystyrene MPs on the immune response and antioxidant system of GIFT (*O. niloticus*).

## 2. Materials and Methods

### 2.1. Experimental Design and Sample Collection

Different sizes of polystyrene microsphere MPs were obtained from Da’e Scientific Co., Ltd. (Tianjin, China). Beads with a diameter of 0.1 mm (excitation: 488 nm, emission: 518 nm) in deionized water, at dispersions (10 mg·mL^−1^, 18,198 × 10^9^ particles mL^−1^), and different experimental concentrations in aquariums were diluted immediately before use in order to prevent agglomeration. The microalga *Chlorella* was cultured with sterile f/2 medium in filtered artificial seawater (artificial sea salt: Institute of Hydrobiology, Chinese Academy of Sciences, Wuhan, China). The cells were grown in a 2 L Erlenmeyer flask at 27 ± 1 °C under cool continuous white fluorescent lighting (5000 lx) with a 12/12 h light/dark cycle and were shaken twice a day to avoid cell precipitation. The inoculum and toxicity test medium and flasks were autoclave-sterilized for 30 min at 121 °C.

This study was conducted at the Freshwater Fisheries Research Center, Chinese Academy of Fishery Sciences, in Wuxi, China. GIFT (*O. niloticus*) juveniles were obtained from a fish farm in Yixing, China. Before the start of the experiments, the fish were acclimated under laboratory conditions. A total of 24 glass aquaria with a capacity of 36 L were used, each containing 30 L of water. The fish were acclimatized for two weeks in de-chlorinated tap water (temperature 29 ± 1 °C; dissolved oxygen > 6 mg L^−1^; pH 6.9–7.1) prior to the experiment. The initial weight of one-year-old fish was 5.1 ± 0.2 g, and the stocking density was 15 fish/36 L. The fish were fed on commercial pellet feeds (Dayu Aquarium Co., Ltd., Guangzhou, China) once a day at a rate of 5% of their body weight. To keep the water oxygenated, air stones were put into each tank. Suction was employed on a daily basis to extract feces and leftover feed. No mortality was observed in the acclimated population.

Then, MPs were added to the corresponding tanks to achieve final exposure concentrations of 0 (group A), 75 nm (group B), 7.5 μm (group C), 750 μm (group D), 75 nm + 7.5 μm + 750 μm (group E), and 75 nm + Chlorella (group F) in 4 replicates for 14 days. The concentration, time point, and duration were in accordance with previous studies [6,20,21]. On the 7th, 10th, and 14th days of the experiment [20,21,22], prior to sampling, fish were feed-deprived for 1 day. Three fish samples were randomly chosen from each aquarium, rinsed with 3% methanol to remove of particles from the skin, anaesthetized with buffered dilute MS−222 (10 mg L^−1^), and dissected by cervical transection. The tissues (brain, gills, liver, and intestine) were extracted, rinsed in 0.15 M KCl, weighed, and subsequently stored at −80 °C to study antioxidant enzymes’ activities.

### 2.2. Antioxidant Status and Immune Response Evaluation

Samples were homogenized in ice with cold 0.86% physiological salt water (1:9, *w*/*v*) and were centrifuged at 5000× r/min at 4 °C for 5 min. The supernatant was collected for the analysis of SOD and ROS content using commercial kits obtained from Nanjing Jiancheng Bioengineering Institute (Nanjing, China). The experiments were conducted following the guidelines provided by the manufacturer [22]. Spectrophotometric quantification was assessed using Power-Wave XS2 (BioTek instruments Inc., Winooski, VT, USA). The WST-1 method was used to measure SOD concentrations through nitro blue tetrazolium inhibition at 450 nm. ROS levels were determined by the oxidation of DHR 123 (dihydrorhodamine 123) to fluorescent rhodamine 123 [23], and without MDA determination according to previous studies [24]. The enzyme activities of IL-1β and TNF-α were determined by diagnostic reagent kits (Comin Biotechnology Co., Ltd., Suzhou, China), and the determination and calculation methods were completed in accordance with previous studies [25,26].

### 2.3. Data Analysis

The Shapiro–Wilk test was used to determine the normality of the data. In cases where the data did not exhibit a homogeneous distribution, a log_2_ transformation was applied. The IBM SPSS statistics software ver. 25 for Windows was used to conduct one-way factorial ANOVA. The significance of treatment was tested by Tukey’s multiple range tests. The results were presented as means ± SEM (standard error of the mean). Statistical significance was determined at a significance level of *p* < 0.05.

## 3. Results

### 3.1. ROS Content

Compared to the control group, there were significant (Table 1, *p* < 0.05) reductions in ROS contents in the fish brain and intestines in all exposed groups. Significant elevation in ROS content was observed in the gills of group B and in the liver of group D (*p* < 0.05). A significant increase in ROS content was observed in the gills and the liver of exposure group C and in the brain of exposure groups C and D with a decrease in all other groups (*p* < 0.05). The intestine exhibited a significant reduction in ROS content across all experimental groups (*p* < 0.05).

The ROS content in the brain and liver exhibited fluctuations, with a significant increase in groups C and D for the brain and groups B/C/D for the liver followed by a reduction in the remaining groups (*p* < 0.05). With the exception of group F, ROS content in the gills increased significantly in all of the treatment groups (*p* < 0.05). The intestine ROS content showed a significant decrease in all treatment groups (*p* < 0.05).

### 3.2. SOD Activities

MPs in all exposed groups markedly reduced SOD activities in the brain and intestine. A significant increase was observed in SOD activity in group B in the gills and in groups B/C in the liver. On the other hand, SOD activities in the brain were significantly reduced in treatment groups E and F compared to other groups, as indicated in Table 2 (*p <* 0.05). A notable rise in enzyme activity was observed in the gills of all groups (*p* < 0.05). The liver showed a significant increase in SOD activity among groups B/C (*p* < 0.05). MPs in all six exposure groups significantly reduced SOD enzyme activity in the intestine (*p* < 0.05).

In terms of SOD activities in the brain, gills, and liver, fluctuations were noted with an increment in treatment groups B/C/D for the brain, groups C/D for the gills and groups B/C for the liver with a reduction in the remaining treatment groups. SOD activities in the intestines significantly reduced across all the treatment groups (*p* < 0.05).

### 3.3. IL-1β Activity

Compared to the control group, a significant (Table 3, *p* < 0.05) elevation of IL-1β levels was recorded in the brain in exposure groups E/F at day 7, in the intestine in exposure groups C/E, and in the gills in group B, with a decrease in all other groups. Significantly, higher levels of IL-1β in the liver were detected in all six exposure groups (*p* < 0.05).

The significant increments of IL-1β levels in the brain and intestine were noted in groups C/D on day 10 and in the liver in groups B/C, while the remaining groups had a reduction (*p* < 0.05). A significant increase in IL-1β activity was observed in the gills except for treatment group D (*p* < 0.05).

Day 14 exposure recorded a significant increase in IL-1β levels in the brain and gills except for group F (*p* < 0.05). Similarly, a significant increase was seen in the liver except for groups E/F, but the intestine showed a significant decrease across all the treatment groups (*p* < 0.05).

### 3.4. TNF-α Activity

TNF-α in the brain and liver was significantly (Table 4, *p* < 0.05) reduced in all MPs-exposed groups. A considerably higher level of TNF-α in the gills was detected in group A, though not statistically different from the control on day 7. TNF-α in the intestine was significantly higher in group C than in all the other groups relative to the control (*p* < 0.05). TNF-α in the brain significantly reduced in all 6 treatment groups but increased in the gills in group C on day 10, in the liver in group B, and in the intestine in group C (*p* < 0.05).

For TNF-α level in the brain, gills, and liver, a significant elevation was discovered in all treatment groups except for group F, with a significant reduction observed in the intestine across all the treatment groups on day 14 (*p* < 0.05).

## 4. Discussion

The results of the present experiment showed a decrease in reactive oxygen species (ROS) content in the brain and gills in treatment group F after the 14 d exposure to MPs. This evidence may imply that when aquatic organisms are stressed by exposure to MPs, they activate their antioxidant defense mechanisms, allowing them to cope with the oxidative stress caused by MPs. Also, it is possible that the absence of oxidative damage in aquatic organisms in group F can be attributed to their non-exposure to MPs. There was no statistically significant increase in ROS generation induced by exposure of *Chlorella* to MPs compared to the control group. These results were in agreement with previous findings [27]. MPs are able to generate ROS in *Chlorella* by restricting the transfer of light and nutrients, as stated in reference [28]. During the exposure periods, ROS in the brain and gills of group C showed an initial increase followed by a decrease. This phenomenon may be attributed to the efficient catalysis of the antioxidant enzyme responsible for catalyzing ROS to a lesser toxic substance, H_2_O_2_, and then into H_2_O and O_2_. Furthermore, after 14 d, ROS in the gills and brain showed a continuous increase in exposure group D when compared to the control. This observation implies that oxidative damage might have been induced by oxidative stress, caused by the inability of ROS catalyzing enzyme to balance the production and elimination of ROS. According to the results, the brain recorded the highest ROS production in treatment group D compared to the gills, liver, and intestine. This finding suggests that exposure to MPs at a concentration of 750 μm for a duration of 14 days resulted in a greater induction of oxidative stress in the brain than the other examined tissues. Previous studies showed that a net with a mesh size of 330 μm was commonly used for sample collection [29], and >500 μm in water and sediments were identified most commonly in India [30]. Throughout the exposure period, the production of ROS in the liver across all the concentrations increased with each passing day, indicating that the liver could have suffered oxidative stress. Apart from the control, it was observed that after the entire exposure period, ROS in the intestine increased initially and then decreased in all the exposure groups except group F, which may indicate that there was enough antioxidant enzyme activity to regulate the excessive production of ROS in the intestine. In group F in the intestine, an increase in ROS content was observed on the 10th and 14th days. There is a potential likelihood that in the intestine, as the days increased, chlorella adsorbed more MPs, thereby activating the over-production of ROS when being ingested by the fish. This finding is in agreement with previous research [17], which stated that all forms of functionalized MPs physically adsorbed on the surface of *Chlorella* result in stress.

SOD converts superoxide anions (O_2_^−^) into H_2_O and O_2_, thereby protecting organisms from over-production of ROS induced by xenobiotics. In this study, significant reduction of SOD activity in the gills was recorded in exposure groups B/F as the duration of exposure increased. Based on the obtained results, it can be inferred that MPs may have suppressed the catalytic capacity of SOD in both the algae and the fish. There is also the potential for insufficient ROS production in the gills of groups B/F to stimulate elevated SOD activities. In comparison to the control group, elevated SOD activity in the brain and gills of GIFT was observed in exposure group D, suggesting a sensitive enzyme response to MPs. Increased SOD activity may also imply the excess generation of ROS caused by MPs exposure as well as the stimulation of the antioxidant systems against oxidative stress. Notably, in this study, SOD activities in the intestines increased in group F. Thus, it is likely that the rise in ROS-related enzyme (SOD) was a compensatory mechanism to decrease the ROS levels in the target cells. Remarkably, in this study, SOD in exposure group C in the intestine and group F in the liver showed the temporal variability of an upsurge followed by a decline. This observation may be interpreted as an indication of an increased level of antioxidants, which serve the purpose of regulating the process of oxygen reduction. SOD activities were reduced once ROS surpassed the antioxidant defense system’s self-clearing threshold, as the impact of antioxidant activation diminished with prolonged exposure. MPs-induced disruptions in antioxidant enzymes have been revealed in a number of aquatic species; including mussels (*Mytilus* spp.) exposed to MPs at a concentration of 32 mg L^−1^ [31], zebrafish (*D. rerio*) subjected to MPs of 5 mm at concentrations of 20, 200, and 2000 ug L^−1^ [7]; and monogonont rotifer (*B. koreanus*) introduced to MPs (0.05, 0.5 and 6 mm) at a concentration of 10 mg L^−1^ [32]. Currently, there is a lack of comprehensive studies examining the effects of MPs on antioxidant-related enzymes in GIFT. This highlights the need for further studies.

Inflammation is a characteristic of the innate immune response. IL-1β is recognized as a pro-inflammatory cytokine, which is a vital facilitator of inflammatory responses and exerts anti-inflammatory activities. In the present study, the activity of IL-1β in the brain at exposure concentrations of 75 nm and 750 μm and in the gills at an exposure concentration of 750 μm was up-regulated after the 14-day exposure period, which suggested a possible increase in inflammation associated with tumor invasiveness in the brain and gills due to their higher rate of bio-accumulation [7]. Group C exhibited an initial rise in IL-1β levels in the brain followed by a decrease as the duration of exposure increased. This finding is consistent with the ‘hormesis effect’, which states that a short duration of exposure to environmental contaminants can promote a stimulatory response in organisms, while a longer exposure time elicits an inhibitory or toxic effect. This suggests that short-term exposure to MPs may induce an immune response in fish as they attempt to adapt to the associated adverse conditions, such as oxidative stress pathways with the increased ROS and SOD. However, the continuous presence of MPs may impede the hypothetic recovery of homeostasis, especially during prolonged exposure. During the 14 d exposure time, IL-1β activity was up regulated in the liver in response to MPs across all the exposure groups. This finding suggests that long-term exposure to MPs in the liver may result in continuous inflammatory response (the increased IL-1β and TNF-α in the present study) by GIFT. Additionally, the relative expression of IL-1β in the brain was significantly highest in treatment group D at the end of the 14-day exposure time. This observed increase may possibly indicate that MPs in group D could result in the overexpression of IL-1β in the GIFT brain, which may be a sign of tumor progression in the brain. MPs are known to prompt immune responses similar to those induced by parasites, such as immune cell recruitment, alterations in immune gene regulation, stress, and immune cell activation.

TNF-α is a pro-inflammatory cytokine which initiates inflammatory processes and intensifies additional inflammatory routes by stimulating other inflammatory molecules. The outcome of this study showed an up-regulation in TNF-α level in the brain at concentrations of groups B/D/E and in the gills at a concentration of 750 μm. This data may postulate an increase in the risk of inflammatory conditions posed by the presence of MPs. Exposure at 75 nm after the 14-day period led to an intriguing result in the gills with an initial increase in TNF-α levels followed by a decrease then an increase. TNF-α expression rises right after inflammation, causing an inflammatory response and then falls. A later elevation in TNF-α expression may be indicative of the onset of infection caused by MPs. Furthermore, the results of this study revealed an up- regulation of TNF-α in the liver among exposure groups B/C/D and in the intestine at 75 nm after the 14-day period. This study postulated that continuous upregulation of TNF-α expression following exposure to MPs could potentially be attributed to heightened inflammation in the affected tissues, leading to potential cellular harm. TNF-α levels in the intestine of group D showed an initial increase, then a decrease, suggesting that probably enough TNF-α was released to remove MPs; hence, infection was reduced. Interestingly, variations were observed in the responses to the same concentration of MPs among the different tissues. The highest TNF-α level was recorded in the brain at an exposure concentration of 750 μm, indicating that MPs may have a greater potential to induce inflammation in the brain compared to the other tissues. Xenobiotics including MPs trigger immune responses in invertebrates, according to previous studies [33]. MPs at 5 μm and 70 nm when exposed to zebrafish had toxic effects and caused inflammation [7]. The fact that MPs exposure could interfere with GIFT’s immune response may have consequences on their ability to respond to parasites, as immune response resources may be depleted leading to increased susceptibility. Further research should be conducted on the effects of MPs on immune related genes in GIFT to bridge the current knowledge gap.

## 5. Conclusions

MPs can induce excessive ROS production and affect the antioxidative enzyme system in GIFT, with the highest observed in the brain at 750 μm. The algae (*Chlorella*) may suffer oxidative stress caused by the presence of MPs. A total of 750 μm MPs can cause the over expression of IL-1β and TNF-α in the brain of GIFT as compared to the other tissues. This observation suggests that MPs can induce the up-regulation of pro-inflammatory cytokines, thereby leading to an inflammation response. This study provides evidence that MPs have the potential to affect the antioxidant system and immune response of GIFT.

## Figures and Tables

**Table 1 biology-12-01430-t001:** ROS content in GIFT exposed to MPs at days 7, 10, and 14.

Tissue	Time (d)	Group A	Group B	Group C	Group D	Group E	Group F
Brain	7	40.82 ± 0.48 ^e^	11.05 ± 0.13 ^a^	17.74 ± 0.21 ^b^	9.61 ± 0.11 ^a^	29.15 ± 0.34 ^c^	34.17 ± 0.40 ^d^
	10	33.54 ± 0.39 ^c^	21.04 ± 0.25 ^b^	59.02 ± 0.69 ^e^	40.14 ± 0.47 ^d^	5.83 ± 0.07 ^a^	22.27 ± 0.26 ^b^
	14	19.80 ± 2.33 ^ab^	21.51 ± 0.25 ^ab^	30.69 ± 0.36 ^b^	508.37 ± 5.99 ^c^	12.12 ± 0.14 ^a^	10.77 ± 0.13 ^a^
Gill	7	61.99 ± 0.73 ^d^	253.46 ± 2.99 ^e^	37.05 ± 0.44 ^b^	21.80 ± 0.26 ^a^	52.63 ± 0.62 ^c^	25.56 ± 0.30 ^a^
	10	39.19 ± 0.46 ^b^	41.31 ± 0.49 ^b^	92.40 ± 1.09 ^d^	26.10 ± 0.31 ^a^	52.19 ± 0.61 ^c^	28.34 ± 0.33 ^a^
	14	36.75 ± 0.43 ^b^	56.56 ± 0.67 ^c^	57.34 ± 0.67 ^c^	117.83 ± 1.39 ^e^	64.82 ± 0.76 ^d^	22.40 ± 0.26 ^a^
Liver	7	11.81 ± 0.14 ^d^	6.72 ± 0.08 ^b^	13.05 ± 0.15 ^e^	14.10 ± 0.17 ^f^	5.13 ± 0.06 ^a^	9.68 ± 0.11 ^c^
	10	15.55 ± 0.18 ^d^	12.65 ± 0.15 ^c^	24.70 ± 0.29 ^e^	12.73 ± 0.15 ^c^	6.48 ± 0.08 ^b^	4.68 ± 0.05 ^a^
	14	21.59 ± 0.25 ^b^	96.61 ± 1.14 ^e^	61.64 ± 0.73 ^d^	27.20 ± 0.32 ^c^	9.32 ± 0.11 ^a^	7.78 ± 0.09 ^a^
Intestine	7	4.77 ± 0.06 ^d^	2.09 ± 0.02 ^a^	4.78 ± 0.06 ^d^	2.05 ± 0.02 ^a^	4.08 ± 0.05 ^c^	3.85 ± 0.04 ^b^
	10	18.76 ± 0.22 ^f^	13.00 ± 0.15 ^c^	15.67 ± 0.18 ^e^	11.71 ± 0.14 ^b^	14.12 ± 0.17 ^d^	8.08 ± 0.09 ^a^
	14	20.71 ± 0.24 ^c^	9.82 ± 0.11 ^b^	6.95 ± 0.08 ^a^	7.30 ± 0.09 ^a^	10.04 ± 0.12 ^b^	9.39 ± 0.11 ^b^

Note: Data are mean values and standard error (±SE); values in the same row with different lowercase letters indicate significant differences among different concentrations of MPs at the same exposure period according to Tukey’s test, with *p* < 0.05 being considered significant. ROS: reactive oxygen species.

**Table 2 biology-12-01430-t002:** SOD activities in GIFT exposed to MPs at days 7, 10, and 14.

Tissue	Time (d)	Group A	Group B	Group C	Group D	Group E	Group F
Brain	7	1.76 ± 0.02 ^e^	1.32 ± 0.01 ^d^	1.10 ± 0.01 ^c^	0.84 ± 0.01 ^b^	1.15 ± 0.01 ^c^	0.56 ± 0.01 ^a^
	10	0.77 ± 0.01 ^c^	1.26 ± 0.01 ^d^	2.30 ± 0.03 ^e^	2.49 ± 0.03 ^f^	0.06 ± 0.01 ^a^	0.26 ± 0.01 ^b^
	14	0.93 ± 0.01 ^b^	1.38 ± 0.02 ^c^	2.73 ± 0.03 ^d^	8.77 ± 0.10 ^e^	0.32 ± 0.01 ^a^	0.40 ± 0.01 ^a^
Gill	7	3.74 ± 0.04 ^c^	9.50 ± 0.11 ^d^	2.00 ± 0.02 ^b^	0.90 ± 0.01 ^a^	1.09 ± 0.01 ^a^	0.95 ± 0.01 ^a^
	10	0.21 ± 0.01 ^a^	1.00 ± 0.01 ^d^	2.25 ± 0.03 ^e^	0.92 ± 0.01 ^c^	0.68 ± 0.01 ^b^	0.73 ± 0.01 ^b^
	14	0.83 ± 0.01 ^c^	0.25 ± 0.01 ^a^	1.68 ± 0.02 ^d^	3.75 ± 0.04 ^e^	0.50 ± 0.01 ^b^	0.45 ± 0.01 ^b^
Liver	7	0.34 ± 0.01 ^c^	0.37 ± 0.01 ^d^	0.57 ± 0.01 ^e^	0.24 ± 0.01 ^b^	0.15 ± 0.01 ^a^	0.14 ± 0.01 ^a^
	10	0.64 ± 0.01 ^d^	1.03 ± 0.01 ^e^	2.31 ± 0.03 ^f^	0.53 ± 0.01 ^c^	0.35 ± 0.01 ^b^	0.28 ± 0.01 ^a^
	14	0.58 ± 0.01 ^b^	4.17 ± 0.05 ^d^	1.38 ± 0.02 ^c^	0.50 ± 0.01 ^b^	0.20 ± 0.01 ^a^	0.18 ± 0.01 ^a^
Intestine	7	0.22 ± 0.01 ^d^	0.07 ± 0.01 ^b^	0.21 ± 0.01 ^d^	0.05 ± 0.01 ^a^	0.14 ± 0.01 ^c^	0.14 ± 0.01 ^c^
	10	0.71 ± 0.01 ^d^	0.37 ± 0.01 ^a^	0.47 ± 0.01 ^b^	0.69 ± 0.01 ^d^	0.63 ± 0.01 ^c^	0.48 ± 0.01 ^b^
	14	1.15 ± 0.01 ^f^	0.56 ± 0.01 ^e^	0.22 ± 0.01 ^b^	0.27 ± 0.01 ^c^	0.13 ± 0.01 ^a^	0.53 ± 0.01 ^d^

Note: Data are mean values and standard error (±SE); values in the same row with different lowercase letters indicate significant differences among different concentrations of MPs at the same exposure period according to Tukey’s test, with *p* < 0.05 being considered significant. SOD: superoxide dismutase.

**Table 3 biology-12-01430-t003:** IL-1β activity in GIFT exposed to MPs at days 7, 10, and 14.

Tissue	Time (d)	Group A	Group B	Group C	Group D	Group E	Group F
Brain	7	0.78 ± 0.01 ^b^	0.76 ± 0.01 ^b^	1.07 ± 0.01 ^c^	0.66 ± 0.01 ^a^	1.11 ± 0.01 ^c^	1.53 ± 0.02 ^d^
	10	1.96 ± 0.02 ^c^	1.43 ± 0.02 ^b^	3.49 ± 0.04 ^e^	2.41 ± 0.03 ^d^	0.59 ± 0.01 ^a^	0.62 ± 0.01 ^a^
	14	0.82 ± 0.01 ^a^	2.63 ± 0.03 ^c^	2.25 ± 0.03 ^c^	17.27 ± 0.20 ^d^	1.53 ± 0.02 ^b^	0.66 ± 0.01 ^a^
Gill	7	2.46 ± 0.03 ^d^	5.33 ± 0.06 ^e^	1.15 ± 0.01 ^c^	0.55 ± 0.06 ^a^	0.87 ± 0.01 ^b^	0.66 ± 0.01 ^a^
	10	0.72 ± 0.01 ^b^	1.18 ± 0.01 ^d^	1.68 ± 0.02 ^e^	0.64 ± 0.01 ^a^	0.79 ± 0.01 ^c^	0.77 ± 0.01 ^bc^
	14	0.97 ± 0.01 ^b^	1.83 ± 0.02 ^e^	1.28 ± 0.01 ^d^	3.35 ± 0.04 ^f^	1.16 ± 0.01 ^c^	0.57 ± 0.01 ^a^
Liver	7	0.28 ± 0.01 ^a^	0.30 ± 0.01 ^a^	0.33 ± 0.01 ^b^	0.39 ± 0.01 ^c^	0.29 ± 0.01 ^a^	0.33 ± 0.01 ^b^
	10	0.90 ± 0.01 ^c^	1.23 ± 0.01 ^d^	1.60 ± 0.02 ^e^	0.54 ± 0.01 ^b^	0.52 ± 0.01 ^ab^	0.48 ± 0.01 ^a^
	14	0.79 ± 0.01 ^c^	3.97 ± 0.05 ^f^	1.97 ± 0.02 ^e^	1.21 ± 0.01 ^d^	0.65 ± 0.01 ^b^	0.51 ± 0.01 ^a^
Intestine	7	0.15 ± 0.01 ^c^	0.05 ± 0.01 ^a^	0.27 ± 0.01 ^e^	0.05 ± 0.01 ^a^	0.18 ± 0.00 ^d^	0.13 ± 0.01 ^b^
	10	0.91 ± 0.01 ^d^	0.83 ± 0.01 ^c^	1.12 ± 0.01 ^e^	1.10 ± 0.01 ^e^	0.59 ± 0.01 ^b^	0.43 ± 0.01 ^a^
	14	0.95 ± 0.01 ^f^	0.65 ± 0.01 ^d^	0.78 ± 0.01 ^e^	0.42 ± 0.01 ^b^	0.33 ± 0.01 ^a^	0.51 ± 0.01 ^c^

Note: Data are mean values and standard error (±SE); values in the same row with different lowercase letters indicate a significant difference among different concentrations of MPs at the same exposure period according to Tukey’s test, with *p* < 0.05 being considered significant.

**Table 4 biology-12-01430-t004:** TNF-**α** activity in GIFT exposed to MPs at days 7, 10, and 14.

Tissue	Time(d)	Group A	Group B	Group C	Group D	Group E	Group F
Brain	7	3.27 ± 0.04 ^d^	1.65 ± 0.02 ^b^	1.29 ± 0.01 ^a^	1.31 ± 0.01 ^a^	1.58 ± 0.02 ^b^	1.99 ± 0.02 ^c^
	10	10.07 ± 0.12 ^f^	3.69 ± 0.04 ^c^	8.84 ± 0.10 ^e^	4.43 ± 0.05 ^d^	2.36 ± 0.03 ^b^	1.25 ± 0.01 ^a^
	14	4.05 ± 0.05 ^b^	10.98 ± 0.13 ^d^	7.58 ± 0.09 ^c^	43.99 ± 0.52 ^e^	5.04 ± 0.06 ^b^	1.87 ± 0.02 ^a^
Gill	7	5.77 ± 0.07 ^e^	5.95 ± 0.07 ^e^	1.48 ± 0.02 ^d^	0.59 ± 0.01 ^a^	1.20 ± 0.01 ^c^	0.96 ± 0.01 ^b^
	10	1.66 ± 0.02 ^e^	1.08 ± 0.01 ^b^	3.66 ± 0.04 ^f^	1.38 ± 0.02 ^d^	1.24 ± 0.01 ^c^	0.97 ± 0.01 ^a^
	14	1.68 ± 0.02 ^b^	2.36 ± 0.03 ^c^	3.17 ± 0.04 ^d^	4.29 ± 0.05 ^e^	2.22 ± 0.03 ^c^	1.00 ± 0.01 ^a^
Liver	7	1.92 ± 0.02 ^d^	1.03 ± 0.01 ^b^	0.95 ± 0.01 ^a^	1.29 ± 0.01 ^c^	0.99 ± 0.01 ^ab^	1.23 ± 0.01 ^c^
	10	3.44 ± 0.04 ^d^	5.02 ± 0.06 ^f^	3.81 ± 0.04 ^e^	1.25 ± 0.01 ^a^	2.28 ± 0.03 ^c^	1.55 ± 0.02 ^b^
	14	3.11 ± 0.04 ^b^	14.30 ± 0.17 ^e^	5.22 ± 0.06 ^d^	4.41 ± 0.05 ^c^	1.22 ± 0.01 ^a^	1.45 ± 0.02 ^a^
Intestine	7	0.36 ± 0.01 ^e^	0.09 ± 0.01 ^b^	0.51 ± 0.01 ^f^	0.08 ± 0.01 ^a^	0.23 ± 0.01 ^d^	0.21 ± 0.00 ^c^
	10	1.79 ± 0.02 ^e^	1.12 ± 0.01 ^c^	2.21 ± 0.03 ^f^	1.46 ± 0.02 ^d^	0.91 ± 0.01 ^b^	0.57 ± 0.01 ^a^
	14	1.91 ± 0.02 ^e^	1.21 ± 0.01 ^c^	1.28 ± 0.01 ^d^	0.69 ± 0.01 ^ab^	0.62 ± 0.01 ^a^	0.69 ± 0.01 ^b^

Note: Data are mean values and standard error (±SE); values in the same row with different lowercase letters indicate significant difference among different concentrations of MPs at the same exposure period according to Tukey’s test, with *p* < 0.05 being considered significant.

## Data Availability

Data are contained within the article.

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
