# Peer review of "Effect of Polystyrene Microplastics on the Antioxidant System and Immune Response in GIFT (Oreochromis niloticus)"

_biology, 2023, doi:10.3390/biology12111430_

Round 1
Reviewer 1 Report
Comments and Suggestions for Authors
Effect of polystyrene microplastics on the antioxidant system 2 and immune response in GIFT (Oreochromis niloticus)
1. The whole manuscript need English editing.
2. Add the list of abbreviation
3. The manuscript need updated references
4. Line 92-93 what the authors mean by different experimental concentrations in aquariums were diluted immediately before use……need more explanation?
5. Line 93…revise this number 18198 109 particles mL-1
6. Line 12 what do you mean by genetically improved farmed tilapia (GIFT)?
7. Mention the dimension of the glass aquaria of 36 L
8. On what base the authors choose the concentration of the experimental MPs? As ell as, the time point and duration of the experiment?
9. The fish rinsed with methanol to get rid of particles from the skin???? Methanol is toxic , what your precaution and doses?
10. anaesthetized with buffered MS-222 , mentioned the method and dose of anesthesia?
11. How the authors examine the data for normality?
12. How the authors examine the sample size ?
13. Write in details about the ROS content analysis as well as, the IL-1ß activity and the TNF- α
14. It is a preliminary study need more experimental analysis on the level of the molecular and histopathological base to be suitable for publication on the Biology journal
Comments on the Quality of English Language
Extensive editing of English language required
Author Response
Title: Effect of polystyrene microplastics on the antioxidant system 2 and immune response in GIFT (Oreochromis niloticus)
- The whole manuscript need English editing.
Response: Thank you for the review’s suggestions. In the Acknowledgments section, the sentence “We thank Teresa (English editors both worked for editage), Ampeire Yona for providing grammar and spelling check of the manuscript.” presented, and the revised version has also been reviewed by the native English speakers (use red track). All the authors hope it will be satisfied by the potential readers.
- Add the list of abbreviation
Response: Thank you for the review’s suggestions. The abbreviation list has been added in line 342-6..
- The manuscript need updated references
Response: Thank you for the review’s suggestions. Some references have been updated in line 363-7, 384-99, 423-5, 434-6, etc.
- Line 92-93 what the authors mean by different experimental concentrations in aquariums were diluted immediately before use……need more explanation?
Response: Thank you for the review’s suggestions, and “for preventing agglomeration” has been added in line 98.
- Line 93…revise this number 18198 109 particles mL-1
Response: Thank you for the review’s suggestions. The number has been revised in line 96.
- Line 12 what do you mean by genetically improved farmed tilapia (GIFT)?
Response: Thank you for the review’s suggestions. The term “genetically improved farmed tilapia” has published more than 180 papers, in 2023 for example, the abstract in the reference (Ye et al., 2023) said that “In the present study, we determined RES administration on these immune tissues transcriptomic response in genetically improved farmed tilapia (GIFT), and further analyzed the relationship between transcriptomic response and intestinal microbiota.”. It is a kind of tilapia especially developed and grown in China. “a kind of tilapia, named as” has been added in line 13.
Reference:
Ye W, Zheng Y, Sun Y, Li Q, Zhu H, Xu G. Transcriptome analysis of the response of four immune related organs of tilapia (Oreochromis niloticus) to the addition of resveratrol in feed. Fish Shellfish Immunol. 2023,133:108510. doi: 10.1016/j.fsi.2022.108510.
- Mention the dimension of the glass aquaria of 36 L
Response: Thank you, the glass aquaria was 36L, and according to the rule of “one g per liter”, may need 15 fish *5.1 g=76.5 L. The further study will be under this rule even though this study may be not satisfy.
- On what base the authors choose the concentration of the experimental MPs? As ell as, the time point and duration of the experiment?
Response: Thank you for the review’s suggestions. The concentration, time point and duration has been followed the references [6] and [20-21], which has been explained in the Materials and Methods section in line 122-3.
- Ding, J.; Zhang, S.; Razanajatovo, R.M.; Zou, H.; Zhu, W. Accumulation, tissue distribution, and biochemical effects of polystyrene microplastics in the freshwater fish red tilapia (Oreochromis niloticus). Environ Pollut. 2018, 238, 1-9. doi:10.1016/j.envpol.2018.03.001.
- Hasan, J.; Siddik, M.A.; Ghosh, A.K.; Mesbah, S.B.; Sadat, M.A.; Shahjahan, M. Increase in temperature increases ingestion and toxicity of polyamide microplastics in Nile tilapia. Chemosphere. 2023, 327, 138502. doi: 10.1016/j.chemosphere.2023.138502.
- Zheng, Y.; Yuan, J.; Gu, Z.; Yang, G.; Li, T.; Chen, J. Transcriptome alterations in female Daphnia (Daphnia magna) exposed to 17β-estradiol. Environ Pollut. 2020, 261, 114208. doi:10.1016/j.envpol.2020.114208.
- The fish rinsed with methanol to get rid of particles from the skin???? Methanol is toxic , what your precaution and doses?
Response: Thank you for the review’s suggestions. The dose “3%” has been added in line 125, and the authors wear the masks to make the precaution.
- anaesthetized with buffered MS-222 , mentioned the method and dose of anesthesia?
Response: Thank you for the review’s suggestions. The dose “(10 mg.L-1)” and “dilute” has been added in line 126.
- How the authors examine the data for normality?
Response: Thank you for the review’s suggestions. The sentence “Shapiro-Wilk test has been used for normality of the data, and a log2 transformation was used for data without a homogeneous distribution.” has been added in line 145-6.
- How the authors examine the sample size ?
Response: Thank you for the review’s suggestions. 15 fish was used for three sampling time points, and fish were randomly selected for sampling, which has been presented in line 124.
- Write in details about the ROS content analysis as well as, the IL-1ß activity and the TNF- α
Response: Thank you for the review’s suggestions. “The experiments were carried out in accordance with the manufacturer’s instructions [26].”, and the calculation method “Briefly, WST-1 method was used to measure SOD concentrations through nitro blue tetrazolium inhibition at 450 nm.” has already presented in line 137-8. “, and the determination methods followed the previous studies [25-26]” has been added for IL-1ß and TNF- α determination in line 142-3.
- It is a preliminary study need more experimental analysis on the level of the molecular and histopathological base to be suitable for publication on the Biology journal
Response: Thank you for the review’s suggestions. The authors appreciate the reviewer's suggestions and also thing it need more test for the mode of action. The data on MPs bioaccumulation in freshwater fish, especially at the tissue level, is still scanty. The general purpose of this study is to evaluate the effects of polystyrene MPs on immune response and antioxidant system of tilapia. The preliminary study only detected some activities in four tissues, it was urgent for us to find the target tissue, and then the further study may focus on the mode of action within a tissue. It will inspire us for the further study when this study can be published in Biology journal.
Reviewer 2 Report
Comments and Suggestions for Authors
Dear authors,
Thank you for your submission. The topic of the effect of microplastic on GIFT and its immune response is impressive.
All methods in this research are sufficient in detail with appropriate statistical tests.
I would recommend a minor revision as follows:
Point 1: What type of plastic is used for MPs in this experiment?
Point 2: The authors did not mention the age of the fish used in this experiment, which is also important.
Point 3: Did the authors observe the MPs in the algae or fish tissue in this study? As the authors mentioned, the data on MPs bioaccumulation in freshwater fish at the tissue level is crucial.
Point 4: Why do 750 μm MPs affect the amount of ROS and the antioxidative enzyme system most in GIFT? Does it relate to the size of the MPs commonly found in the environment? How about other research?
Point 5: All Tables' format needs to be revised to follow.
Point 6: The authors could explain more about the effect of MPs against different parts of GIFT tissues and the relationships between SOD activity, IL-1ß activity, and TNF-α activity in each tissue.
Please check the scientific name throughout the manuscript, for example:
Line 78: Chlorella must be italicized, not sp
Line 217, 220, 315: Chlorella
Author Response
Dear authors, Thank you for your submission. The topic of the effect of microplastic on GIFT and its immune response is impressive. All methods in this research are sufficient in detail with appropriate statistical tests. I would recommend a minor revision as follows:
Point 1: What type of plastic is used for MPs in this experiment?
Response: Thank you for the review’s suggestions. “polystyrene” has already in the introduction in line 90, and “polystyrene microsphere” has been added in the in the abstract and Materials and Methods sections in line 11-2, 94.
Point 2: The authors did not mention the age of the fish used in this experiment, which is also important.
Response: Thank you for the review’s suggestions. “one-year old” has been added in line 111.
Point 3: Did the authors observe the MPs in the algae or fish tissue in this study? As the authors mentioned, the data on MPs bioaccumulation in freshwater fish at the tissue level is crucial.
Response: Thank you for the review’s suggestions. The sentence “, and the most important thing is the non-ignorable effects on the health and metabolism of fish tissues.” has been added in line 88-9. The present study only detect the four kinds of the enzymic activities in four tissues, and the detection of MPs in the fish tissue or algae has not been performed. The further study will focus on this until this work has been accepted by the Biology journal.
Point 4: Why do 750 μm MPs affect the amount of ROS and the antioxidative enzyme system most in GIFT? Does it relate to the size of the MPs commonly found in the environment? How about other research?
Response: Thank you for the review’s suggestions. Brain SOD/ROS and gill SOD, brain IL-1ß and TNF-α has significantly increased in 750 μm MPs groups, and the sentence “The brain recorded the highest ROS production in treatment group D than in the gills, liver and intestine suggesting that MPs at concentration 750 μm after the 14-d exposure period was able to cause oxidative stress more in the brain than the other tissues.” Presented in line 236-9.
The sentence “Previous studies showed that 330 μm was the typical mesh size when we used a net for sample collection [29], and >500 μm in water and sediments were identified most commonly in India [30]” has been added in line 239-41.
Reference:
- Li, J.; Liu, H.; Paul Chen, J. Microplastics in freshwater systems: A review on occurrence, environmental effects, and methods for microplastics detection. Water Res. 2018, 137, 362-374. doi: 10.1016/j.watres.2017.12.056.
- Neelavannan, K.; Sen, I.S. Microplastics in freshwater ecosystems of India: current trends and future perspectives. ACS Omega 2023, 8, 34235-34248. doi: 10.1021/acsomega.3c01214.
Point 5: All Tables' format needs to be revised to follow.
Response: Thank you for the review’s suggestions. The Tables' format has been revised in line 195-204.
Point 6: The authors could explain more about the effect of MPs against different parts of GIFT tissues and the relationships between SOD activity, IL-1ß activity, and TNF-α activity in each tissue.
Response: Thank you for the review’s suggestions. The sentence “because its higher bio-accumulation rate [7]…(like oxidative stress pathway with the increased ROS and SOD)…(the increased IL-1ß and TNF-α in the present study)” has been added in line 286, 292-3, 298.
Please check the scientific name throughout the manuscript, for example:
Line 78: Chlorella must be italicized, not sp, Line 217, 220, 315: Chlorella
Response: Thank you for the review’s suggestions, corrected as suggested in line 80, 228, 252, 336.

Reviewer 3 Report
Comments and Suggestions for Authors
Line 35: Reframe the sentence
Line 59: Recheck the line, incomplete sentence
Line 72-76: It would be more appropriate to mention a direct link between the mentioned microalgae and MPs in the initial phrases with reference.
I also have a few queries to be asked to the authors. If possible, I suggest to include the same to increase the merit of the manuscript.
How ROS content has been evaluated. There is no mention of ROS assay in methods.
Why MDA has not been considered for Anti-oxidant assay.
I would suggest performing a molecular expression assay to confirm the enzymatic findings. The inclusion of more parameters will further strengthen the findings.
Comments on the Quality of English Language
Extensive Language editing need to be done
Author Response
Line 35: Reframe the sentence
Response: Thank you for the review’s suggestions, corrected as suggested in line 35-6.
Line 59: Recheck the line, incomplete sentence
Response: Thank you for the review’s suggestions, corrected as suggested in line 59-60.
Line 72-76: It would be more appropriate to mention a direct link between the mentioned microalgae and MPs in the initial phrases with reference.
Response: Thank you for the review’s suggestions, the citation has been added in line 75.
I also have a few queries to be asked to the authors. If possible, I suggest to include the same to increase the merit of the manuscript. How ROS content has been evaluated. There is no mention of ROS assay in methods.
Response: Thank you for the review’s suggestions, and “ROS levels were determined by the oxidation of DHR 123 (dihydrorhodamine 123) to fluorescent rhodamine 123 [23]” has been added in line 138-40.
Reference:
- Istomina, A.A.; Zhukovskaya, A.F.; Mazeika, A.N.; Barsova, E.A.; Chelomin, V.P.; Mazur, M.A.; Elovskaya, O.A.; Mazur, A.A.; Dovzhenko, N.V.; Fedorets, Y.V.; Karpenko, A.A. The relationship between lifespan of marine bivalves and their fatty acids of mitochondria lipids. Biology (Basel) 2023, 12, 837. doi: 10.3390/biology12060837.
Why MDA has not been considered for Anti-oxidant assay.
Response: Thank you for the review’s suggestions, the MDA (need to, pls. see the references) has not been used for test in fish tissues, and it will be used in the further study. And also some published papers (Li et al., 2022; Wu et al., 2022) only test ROS, SOD, but not MDA. The sentence “and without MDA determination following previous studies [24]” has been added in line 140.
Reference:
Li L, Gao M, Yang N, Ai L, Guo L, Xue X, Sheng Z. Trimethyltin chloride induces apoptosis and DNA damage via ROS/NF-κB in grass carp liver cells causing immune dysfunction. Fish Shellfish Immunol. 2023,142:109082. doi: 10.1016/j.fsi.2023.109082.
Wu L, Dong B, Chen Q, Wang Y, Han D, Zhu X, Liu H, Zhang Z, Yang Y, Xie S, Jin J. Effects of Curcumin on Oxidative Stress and Ferroptosis in Acute Ammonia Stress-Induced Liver Injury in Gibel Carp (Carassius gibelio). Int J Mol Sci. 2023,24(7):6441. doi: 10.3390/ijms24076441.
Wang J, Sun L, Li X, Tao S, Wang F, Shi Y, Guan H, Yang Y, Zhao Z. Alkali exposure induces autophagy through activation of the MAPKpathway by ROS and inhibition of mTOR in Eriocheir sinensis. Aquat Toxicol. 2023,258:106481. doi: 10.1016/j.aquatox.2023.106481.
- Li A, Gu Y, Zhang X, Yu H, Liu D, Pang Q. Betaine Regulates the Production of Reactive Oxygen Species Through Wnt10b Signaling in the Liver of Zebrafish. Front Physiol. 2022,13:877178. doi: 10.3389/fphys.2022.877178.
Wu T, Fan T, Xie Y. Antagonism of Cyanamide-3-O-glucoside and protocatechuic acid on Aflatoxin B1-induced toxicity in zebrafish larva (Danio rerio). Toxicon. 2022,216:139-147. doi: 10.1016/j.toxicon.2022.06.009.
I would suggest performing a molecular expression assay to confirm the enzymatic findings. The inclusion of more parameters will further strengthen the findings.
Response: Thank you for the review’s suggestions. The authors appreciate the reviewer's suggestions and also thing it need more molecular test for the mode of action. The data on MPs bioaccumulation in freshwater fish, especially at the tissue level, is still scanty. The general purpose of this study is to evaluate the effects of polystyrene MPs on immune response and antioxidant system of tilapia. The preliminary study only detected some activities in four tissues, it was urgent for us to find the target tissue, and then the further study (especially the molecular test) may focus on the mode of action within a tissue. It will inspire us for the further study when this study can be published in Biology journal.

Round 2
Reviewer 1 Report
Comments and Suggestions for Authors
Accept after minor editing of English language
Comments on the Quality of English LanguageMinor editing of English language required
Author Response
Accept after minor editing of English language, Minor editing of English language required.
Response: Thank you, corrected as suggested in line 15-18, 21-39, 43-47, etc.
Reviewer 3 Report
Comments and Suggestions for Authors
I agree with the explanation but still will highly recommend molecular studies for confirmation of test results.
Comments on the Quality of English LanguageThe authors have sufficiently explained to the queries raised. But language editing from professional or native speakers will improve the MS.
Author Response
I agree with the explanation but still will highly recommend molecular studies for confirmation of test results.
Response: Thank you for the understanding. The present study aimed to find the target tissue involved in the response of different sizes of MPs, and then the further study (especially the molecular test) will be performed in the near future.
The authors have sufficiently explained to the queries raised. But language editing from professional or native speakers will improve the MS.
Response: Thank you, corrected as suggested in line 43-7 (section Introduction), 135-7 (section Materials and Methods), 183 (section Results), 252 (section Discussion), etc.